# Embryogenic Stem Cell Identity after Protoplast Isolation from *Daucus carota* and Recovery of Regeneration Ability through Protoplast Culture

**DOI:** 10.3390/ijms231911556

**Published:** 2022-09-30

**Authors:** Jong-Eun Han, Han-Sol Lee, Hyoshin Lee, Hyunwoo Cho, So-Young Park

**Affiliations:** 1Department of Horticultural Science, Division of Animal, Horticultural and Food Sciences, Chungbuk National University, Cheongju 28644, Korea; 2Department of Forest Genetic Resources, National Institute of Forest Science, 39 Onjeong-ro, Suwon 16631, Korea; 3Department of Industrial Plant Science and Technology, College of Agricultural, Life and Environmental Sciences, Chungbuk National University, Cheongju 28644, Korea

**Keywords:** embryogenic callus-derived protoplast, gene expression, plant regeneration

## Abstract

Protoplasts are single cells isolated from tissues or organs and are considered a suitable system for cell studies in plants. Embryogenic cells are totipotent stem cells, but their regeneration ability decreases or becomes lost altogether with extension of the culture period. In this study, we isolated and cultured EC-derived protoplasts (EC-pts) from carrots and compared them with non-EC-derived protoplasts (NEC-pts) with respect to their totipotency. The protoplast isolation conditions were optimized, and the EC-pts and NEC-pts were characterized by their cell size and types. Both types of protoplasts were then embedded using the alginate layer (TAL) method, and the resulting EC-pt-TALs and NEC-pt-TALs were cultured for further regeneration. The expression of the EC-specific genes *SERK1, WUS, BBM, LEC1,* and *DRN* was analyzed to confirm whether EC identity was maintained after protoplast isolation. The protoplast isolation efficiency for EC-pts was 2.4-fold higher than for NEC-pts (3.5 × 10^6^ protoplasts·g^−1^ FW). In the EC-pt group, protoplasts < 20 µm accounted for 58% of the total protoplasts, whereas in the NEC-pt group, small protoplasts accounted for only 26%. In protoplast culture, the number of protoplasts that divided was 2.6-fold higher for EC-pts than for NEC-pts (7.7 × 10^4^ protoplasts·g^−1^ FW), with a high number of plants regenerated for EC-pt-TALs, whereas no plants were induced by NEC-pt-TAL. Five times more plants were regenerated from EC-pts than from ECs. Regarding the expression of EC-specific genes, *WUS* and *SERK1* expression increased 12-fold, and *LEC1* and *BBM* expression increased 3.6–6.4-fold in isolated protoplasts compared with ECs prior to protoplast isolation (control). These results reveal that the protoplast isolation process did not affect the embryogenic cell identity; rather, it increased the plant regeneration rate, confirming that EC-derived protoplast culture may be an efficient system for increasing the regeneration ability of old EC cultures through the elimination of old and inactivate cells. EC-derived protoplasts may also represent an efficient single-cell system for application in new breeding technologies such as genome editing.

## 1. Introduction

Plant stem cells (PSCs) are undifferentiated cells that develop into different cell types or tissues, undergo uniform cell division, and maintain their cell identity [1]. In plants, PSCs include pluripotent stem cells in the apical meristem and cambium layer in addition to embryogenic stem cells (ESCs), which are totipotent cells induced by in vitro culture conditions [2]. The totipotency of an embryogenic callus (EC) can decrease due to various factors such as cell aging, the number of passages, and exposure to growth regulators during long-term culture. In some cases, regeneration into abnormal plants occurs due to somaclonal variations [3]. Several studies have reported that these problems in EC culture can be overcome using various culture methods or through cryopreservation [4,5].

Among these various culture methods, EC-derived protoplast culture has been used in new breeding techniques such as protoplast fusion and genome editing, and in plant cell research based on single-cell analysis [6,7,8]. The use of EC-derived protoplast culture has been reported for various species such as *Rosa hybrida* [9], *Vitis vinifera* [10], *Crocus cancellatus* [11] *Oryza sativa* [12], and *Cinnamomum camphora* [13], among others.

In order to study the totipotency of EC, many transcriptomic, metabolomic, and proteomic analyses have compared EC with non-embryogenic callus (NEC) [14,15,16,17,18].

Jha et al. [19] reported that various transcription factors (TFs), such as *SOMATIC EMBRYOGENESIS RECEPTOR-LIKE KNASE (SERK)*, *WUSCHEL (WUS)*, *BABY BOOM (BBM), LEAFY COTYLEDON1/2 (LEC1/2), FUSCA3 (FUS3)*, and *ABAINSENSITIVE3 (ABI3)*, are regulated by appropriate expression during the transition from vegetative to EC or the maintenance of EC identity.

The processes of protoplast isolation, protoplast culture, and plant regeneration are useful for studies applying ECs. Therefore, suitable conditions for high-yielding protoplast isolation are required, and the efficiency of this procedure is affected by factors such as the combination of enzyme mixtures and digestion time [20]. We previously conducted protoplast isolation using a variety of plant species and found that the optimal isolation conditions differed for each species [21]. In addition, even within the same species, the optimal digestion time was dependent on the characteristics of the explant. For instance, for the Persian silk tree, 6 h digestion was optimal when using the leaves, whereas 20 h was required for the callus [8].

In this study, we established optimal conditions for protoplast isolation by adjusting the enzyme concentration and immersion time of the digestion solution. We then characterize the cellular and molecular aspects of ECs using EC-derived protoplast culture. In addition, we studied the recovery of regeneration abilities to address the decrease observed in long-term culture with respect to maintenance of ESC identity after protoplast isolation.

## 2. Results and Discussion

### 2.1. Optimization of Protoplast Isolation

The digestion solution was treated with different Celluclast^®^ 1.5 L concentrations and different digestion times. The number of isolated protoplasts in the Celluclast^®^ 1.5 L 3.0% treatment was 12.8 × 10^6^ protoplasts∙g^−1^ FW, higher than 0.5–1.0% (4.4 × 10^6^ protoplasts∙g^−1^ FW) and 2.0% (7.7 × 10^6^ protoplasts∙g^−1^ FW) (Figure 1A). The optimal digestion times differed, with 12 and 14 h resulting in 10.9 × 10^6^ and 11.3 × 10^6^ protoplasts∙g^−1^ FW, respectively, showing 3.0–3.1-fold higher protoplast isolation efficiency compared to 10 h. These results indicate that 3% Celluclast^®^ 1.5 L and 12 h digestion time are optimal for protoplast isolation.

The plant cell wall consists of polysaccharide polymers such as cellulose, hemicellulose, pectin, glycoproteins, and lignin [22]. Cellulose, the main component of the cell wall, has a crystalline structure and is resistant to hydrolysis. Hemicellulose is reported to have an irregular amorphous structure and is easily hydrolyzed by acids/bases or enzymes such as hemicellulose [23,24].

Therefore, cellulose digestion for protoplast isolation is especially important. The digestion solution used in our study contains Viscozyme^®^ and Celluclast^®^ 1.5 L, which contains cellulase, which is the main component (Appendix A). In this study, 3% Celluclast^®^ 1.5 L was the most effective of the digestion solutions tested for protoplast isolation from carrot callus.

In addition to the enzyme concentration in the digestion solution, one of the important factors in protoplast isolation is the time for the digestion of the plant materials. The composition and structure of plant cell walls differ according to species, cell types, explants, developmental stage, etc. [25,26]. Moghaddam and Wilman [27] reported that the thickness of the leaf cell wall ranges from 0.15 to 1.46 μm in eight different species of plants. The different plant materials have a distinct digestion time that is appropriate for cell wall digestion. Leaf-derived protoplast isolation was performed depending on plant species, and the leaves were sufficiently degraded after a digestion time of 3–6 h [7,28,29,30]. By contrast, callus-derived protoplast isolation requires a long digestion time of 12–24 h [7,31,32,33]. Bartos et al. [34] reported that the cell wall thickness was 1.9 and 3.1 μm on the 30th and 150th days of coffee callus culture, respectively.

In this study, the isolation of protoplasts from carrot ECs for 12 h of immersion in enzyme digestion solution was effective. By contrast, Godel-Jędrychowska et al. [6] reported that 16 h immersion in enzyme solution is suitable for protoplast isolation from carrot leaves. Direct comparison is difficult because the type and concentration of the enzyme solution used by Godel-Jędrychowska et al. [6] differed from those used in this study.

### 2.2. Isolation of Protoplasts from EC and NEC

The isolation efficiency for EC-pts was 8.3 × 10^6^ protoplasts∙g^−1^ FW, which was 2.4-fold higher than for NEC-pts (Figure 2A). The EC-pt group comprised 58% of protoplasts < 20 µm, whereas this was the case in only 26% of the NEC-pt group (Figure 2B). Unlike EC-pt, the NEC-pt group comprised 59% of protoplasts > 20 µm and 15% of xylem cells. Additionally, when vacuoles from NEC-pts and EC-pts were stained with neutral red, large cells with single vacuoles were observed in NEC-pts, and small cells with small vacuoles were found in EC-pts (Figure 2C).

The protoplast isolation efficiency of EC was 2.6-fold higher than for NEC because EC experienced lower cell aggregation than NEC; therefore, it is presumed that the digestion solution can easily penetrate cells during protoplast isolation. It was reported that EC has a thin cell wall, and cell aggregation was reduced compared to NEC during cell suspension culture [22,35]. In addition, the callus of EC and NEC induced from coffee leaves were compared; the NEC cell wall was 2.1–3.9-fold thicker than the EC cell wall [34].

EC has the characteristics of a stem cell [2], with small cell size and small vacuoles that are abundantly distributed in the cell [36]. Protoplasts with stem cell characteristics were mainly distributed in the EC-pt group. In the NEC-pt group, we mainly found large-sized protoplasts with developed vacuoles and tracheids, which are xylem vessels.

### 2.3. Regeneration of EC and NEC-Derived Protoplasts

Isolated protoplasts from EC (EC-pt-TAL) and NEC (NEC-pt-TAL) were embedded using the TAL method at 1 × 10^6^ protoplast·mL^−1^ and cultured for 5 weeks. In both cultures, the first cell division event was similarly observed on the fifth day of culture (Figure 3 and Figure 4). Around 2.6-fold more EC-pt-TAL protoplasts (20 × 10^4^ protoplasts∙g^−1^ FW) underwent cell division than NEC-pt-TAL protoplasts (Figure 3 and Figure 4). The second and third divisions were observed from the tenth day. EC-TAL (50.6 × 10^4^ protoplasts∙g^−1^ FW) formed multi-cells 1.4-fold more than NEC-TAL after 21 days (Figure 3). However, after 28 days of culture, micro-calluses were similarly formed in both NEC-TAL and EC-TAL.

EC-pt had more cell division from the early stage of culture than NEC-pt in this study. EC is an embryonic stem cell with morphological and functional characteristics of stem cells [2,36,37,38], and stem cells have continuous cell division [39]. Therefore, we interpreted that carrot EC-pt showed higher first cell division and multi-cells than NEC-pt.

After being transferred to a hormone-free MS medium, both cultures had higher proliferation of micro-calli. After 2 weeks in the medium, EC-pt-TAL showed a globular-shaped structure on the surface of TAL, while callus was seldom observed on NEC-pt-TAL. After eight weeks of culture in a free-hormone medium, young plants developed from EC-pt-TAL via somatic embryogenesis. There were 351 plantlets∙TAL^−1^ regenerated plants from one EC-pt-TAL. However, in NEC-pt-TAL, only callus proliferation was observed without plant regeneration (Figure 5). Only callus formation was observed in NEC-pt without organogenesis or plant regeneration in roses. Whole plants regenerated from EC-pt [9]. *Nigella damascena* had a different regeneration capacity according to the use of different sources for protoplast preparation, such as hypocotyl and cotyledon [32].

### 2.4. Regeneration Capacity of EC-pt-TAL and EC-Specific Gene Expression

To compare the regeneration capacity of EC with that of EC-pt, EC-pt and EC were embedded in TAL and cultured in a liquid medium. The EC-pt-TAL showed fivefold higher plant regeneration than EC-TAL, which had 68 plantlets∙TAL^−1^ of plant regeneration efficiency.

RT-qPCR analysis of EC-specific genes was conducted to examine changes in EC-specific gene profiles before and after protoplast isolation. *SERK1* expression increased 12-fold after protoplast isolation compared to EC before isolation (Figure 6). *WUS* had a similar expression pattern to *SERK1* that increased 12-fold after protoplast isolation and was maintained at 7–8-fold higher than EC for 12–24 h (Figure 6). *BBM* and *LEC1* also increased 6.4-fold and 3.6-fold, respectively, after protoplast isolation compared to EC (Figure 6). Unlike with the other EC-specific genes, no significant differences in gene expression were observed for *DRN* after protoplast isolation (Figure 6).

Since the roles and functions of EC-specific genes differ depending on EC formation or the somatic embryogenesis process, the expression level of each gene may vary to different extents. Xu et al. [40] reported that protoplast isolation significantly increased chromatin accessibility in *Arabidopsis* leaves, resulting in a more than 5-fold increase in ectopic gene expression. *SERK1* is known to be involved in EC formation and expressed at the early globular stage of somatic embryogenesis [41], and *WUS* is also known to encode a transcription factor characteristically expressed in EC formation and maintenance of stem cell identity [42]. In this study, *WUS* and *SERK1* were significantly increased in the early stage after protoplast isolation, and gene expression was maintained over time. The LEC1 is expressed in the early stage of somatic embryogenesis, and BBM occurred up from the globular to the torpedo stage, and both of them are involved in EC development and maintenance of EC identity, and even *BBM* is involved in cell wall biosynthesis [43,44] and showed relatively low gene expression compared with *WUS* and *SERK1* in the early stage of cell division after protoplast isolation. Kusnandar et al. [45] reported that *DRN* expression decreases immediately after protoplast isolation and increases during the active cell division stage in *Arabidopsis* mesophyll cells.

As a result, we found that the expression of *WUS* and *SERK1* significantly increased immediately after protoplast isolation; *LEC1* and *BBM* also increased. These results indirectly demonstrate two important concepts: (1) the embryogenic identity and totipotency were maintained during the isolation process from ECs into a single protoplast, where even the protoplasts were exposed to excessive stress; (2) in the process of single protoplasts regaining their cell division ability, non-embryogenic cells or senescent cells may have been eliminated through bursting or failure to continue cell division. Based on the gene expression results, our hypothesis is supported by the significantly higher number of plants regenerated from EC-derived protoplast culture (EC-pt-TAL) compared with the same amount of EC (EC-TAL).

## 3. Materials and Methods

### 3.1. Plant Materials

Carrot (*Daucus carota* cv. Hong-gwang) seeds were surface sterilized by incubating in 70% EtOH for 30 s, then in 2% (*v*/*v*) sodium hypochlorite containing Tween 20 for 10 min, followed by washing five times with sterilized water. Sterilized seeds were germinated on solid MS medium [46] containing 1.0 mg∙L^−1^ 2,4-D, 30 g∙L^−1^ sucrose, and 2.6 g∙L^−1^ gelrite to induce embryogenic callus, and non-embryogenic callus. The pH of the medium was adjusted to 5.7–5.8 before autoclaving. After 4–6 weeks of culture, EC and NEC were induced from the seeds and subcultured in the same medium every 4 weeks. Induced EC and NEC were cultured in the same medium every 4 weeks at 24 ± 1 °C in the dark.

### 3.2. Optimization of Protoplast Isolation

Protoplast isolation was performed using 300 mg of carrot callus with addition of 3 mL digestion solution (Appendix A), then immersed into a 55 rpm shaker for 12 h. To find the Celluclast^®^ 1.5 L (Novozymes, Bagsværd, Denmark) concentration for cell wall digestion, various concentrations of Celluclast^®^ 1.5 L were investigated ranging 0–3% (*v*/*v*) in the cell wall digestion solution. For effective protoplast isolation, digestion times of 10–14 h were tested using a digestion solution containing 3% Celluclast^®^ 1.5 L. After digestion, the protoplast solution with removed cell wall was filtered through a 100 μm nylon sieve (SPL, Pocheon, Korea) to remove debris and centrifuged at 100× *g* for 5 min, followed by discarding of the supernatant and retention of the protoplast pellet. The protoplast pellet was gently resuspended in washing solution (W5 solution, Appendix A) via pipetting and centrifuged at 100× *g* for 5 min. That washing step was repeated twice. After washing, the protoplast pellet was resuspended in 1 mL of protoplast culture medium (PCM, Appendix A). Protoplast isolation efficiency was determined using a hemocytometer (Sigma-Aldrich, St. Louis, MO, USA) loaded with 10 µL of resuspended protoplasts, and the protoplasts were counted under an optical microscope (DMi8, Leica, Wetzlar, Germany).

### 3.3. EC- and NEC-Derived Protoplast Isolation and Culture

#### 3.3.1. Protoplast Isolation

Following the above method, we conducted the protoplast isolation from EC and NEC by immersion in a digestion solution (Appendix A) for 12 h. After protoplast isolation, we counted the number of protoplasts and measured protoplast size through optical microscopy. The protoplast solution was diluted to 2.0 × 10^6^/mL using PCM for use in protoplast culture.

#### 3.3.2. Protoplast Culture Using Thin Alginate Layer (TAL) System

The protoplast solution isolated from EC and NEC was embedded in culture using a thin alginate layer (TAL) system, according to Jeong et al. [47]. We mixed 1 mL of protoplast solution (2.0 × 10^6^/mL) with an equal volume of sodium alginate solution (Appendix A). Then, 2 mL of the protoplast–alginate mixture was spread onto CaCl_2_ agar (Appendix A) in a 60 mm Petri dish. After 1 h, we added the CaCl_2_ solution (Appendix A) to the upper side of TAL for 30 min for fixation. Immobilized TAL was uniformly divided into 16 equal fragments (1.5 cm^2^), which were transferred into a 16-well-plate with the addition of 1 mL PCM. Protoplasts embedded in TAL were cultured in the dark at 24 ± 1 °C for 4 weeks.

#### 3.3.3. Plant Regeneration from EC-TAL, EC-pt-TAL, and NEC-pt-TAL

EC-pt-TAL and NEC-pt-TAL fragments cultured for 4 weeks were washed in liquid hormone-free MS medium twice and then transferred into plant regeneration medium (PRM). The plates were cultured for 2 weeks in the dark during the developmental stage of EC. Afterward, the plant regenerated from TAL fragments for 4 weeks at 24 ± 1 °C under light conditions. In addition, we compared EC-pt-TAL and EC-TAL to determine the effectiveness of plant regeneration from protoplasts. The EC-TAL was prepared using the above TAL method and mixing of EC with sodium alginate solution. Afterward, fragments of EC-TAL were cultured in PRM for 2 weeks in the dark and 4 weeks in the light.

#### 3.3.4. Microscopy and Staining

After protoplast isolation, the protoplast solution was double stained with 0.005% propidium iodide (PI) and 0.01% fluorescein diacetate (FDA) for 5 min. The living (green) and dead (red) protoplasts were visualized and imaged using fluorescence microscopy (Dmi8, Leica, Wetzlar, Germany). The protoplasts were counted under a microscope, and their diameter was measured using a Leica Application Suite X 6.4v program to investigate protoplast size. The protoplast solution was stained with 0.01% neutral red (Sigma-Aldrich, St. Louis, MO, USA) for 5 min and washed twice with PCM for vacuole staining [48]. The washed protoplasts were observed under light microscopy.

#### 3.3.5. EC-Specific Gene Expression

Samples were prepared for NEC and EC and 0, 12, and 24 h after EC protoplast isolation. Gene expression analysis was performed based on RT-qPCR of total RNA extracted from samples. Embryogenic-specific primer sets were designed and diluted to 10 pmol with tertiary sterile water (Appendix A). Total RNA was isolated using an AccuPrep Universal RNA Extraction Kit (BIONEER Corp., Daejeon, Korea), and cDNA was synthesized using ReverTra Ace qPCR RT Master Mix (Toyobo, Osaka, Japan). RT-qPCR was performed using synthesized cDNA and TB Green PremMix EX Taq II (Takara, Osaka, Japan). RT-qPCR amplification involved 40 cycles at 95 °C for 1 min, 95 °C for 15 s, 56 °C for 30 s, and 72 °C for 30 s. The procedures were performed using a CFX96 Real-Time System (Bio-Rad, Hercules, CA, USA).

#### 3.3.6. Statistical Analysis

The results are presented as mean values and standard errors. One-way analysis of variance (ANOVA) was used to determine whether the groups differed significantly. Statistical assessments of the difference between mean values were then assessed using Duncan’s multiple range test (DMRT) and t-test. A value of *p* = 0.05 was considered to indicate statistically significant differences. All data were analyzed using a SAS program (Software Version 9.4; SAS Institute, Cary, NC, USA).

## 4. Conclusions

In this study, we isolated protoplasts from two different cell types, EC and NEC. The EC-pt group comprised small protoplasts characteristic of embryogenic stem cells (ESCs) and was more efficient in protoplast isolation than NEC-pt. The EC-pt had more divided cells than NEC-pt and successfully regenerated intact plants around 3 months after its preparation. After protoplast isolation, we confirmed that EC-specific genes significantly increased in expression during the early stage of culture. Thus, EC-pts maintain their identity as ESCs after protoplast isolation, and their ability to regenerate is recovered during protoplast culture.

## Figures and Tables

**Figure 1 ijms-23-11556-f001:**
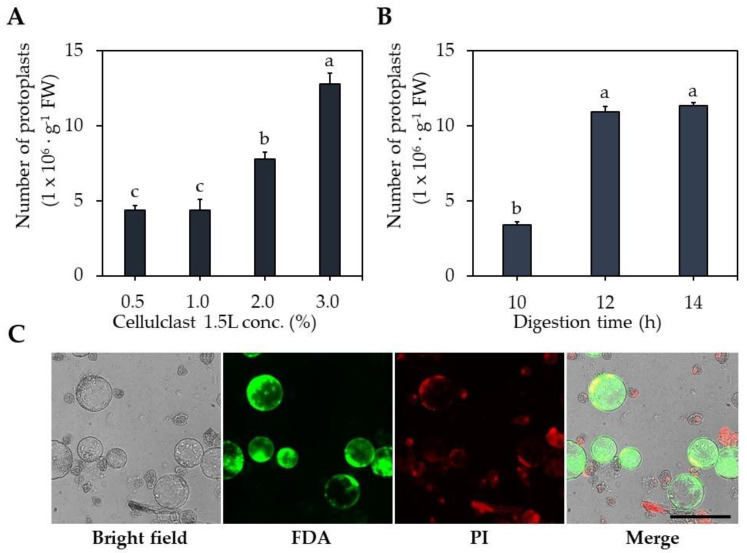
Optimization of carrot protoplast isolation. (**A**) Effect of Celluclast^®^ 1.5 L concentrations. (**B**) Effect of digestion time. (**C**) Protoplast stained with FDA (live cell, green) and PI (dead cell, red). Black scale bar = 100 µm. Different lowercase letters indicate statistically significant differences (*p* < 0.05; Duncan’s multiple range test). Values are expressed as mean ± SD, the error bars represent at least three independent replicates.

**Figure 2 ijms-23-11556-f002:**
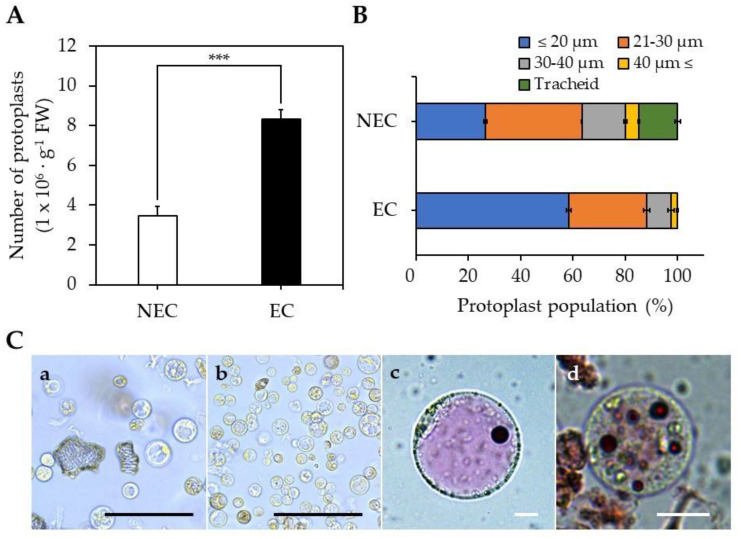
Protoplast isolation from EC and NEC: (**A**) number of isolated protoplasts; (**B**) population of isolated protoplasts by size; (**C**) Protoplasts images of the (**a**) NEC-derived protoplasts and (**b**) EC-derived protoplasts, (**c**) NEC-vacuoles stained with neutral red, (**d**) EC-vacuoles stained with neutral red. Black scale bar = 100 µm, White scale bar = 10 µm. The asterisk indicates statistically significant differences (***, *p* < 0.001; *t*-test). Values are expressed as mean ± SD, the error bars represent at least three independent replicates.

**Figure 3 ijms-23-11556-f003:**
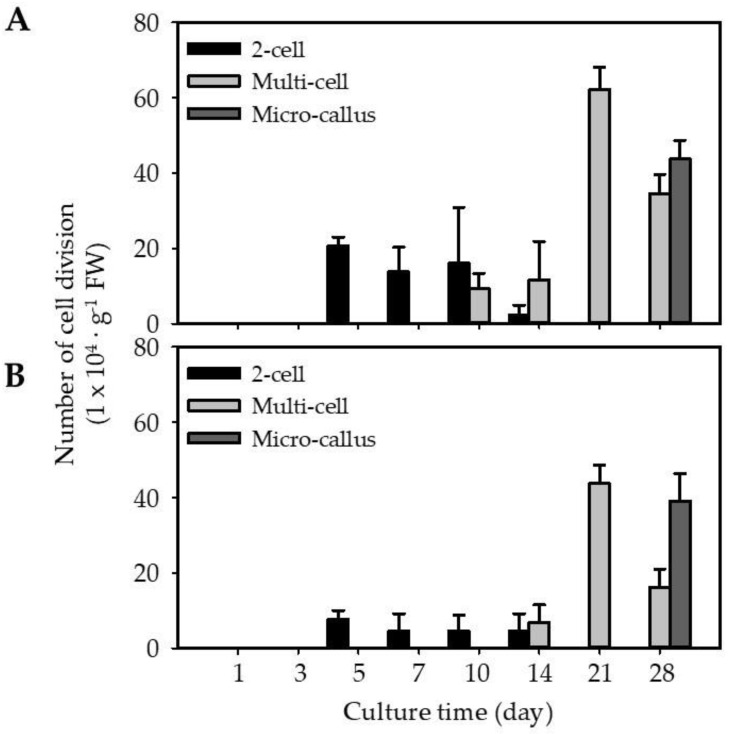
Embedded protoplasts in culture for 4 weeks: (**A**) EC-derived protoplasts; (**B**) NEC-derived protoplasts. Values are expressed as mean ± SD. The error bars represent at least three independent replicates.

**Figure 4 ijms-23-11556-f004:**
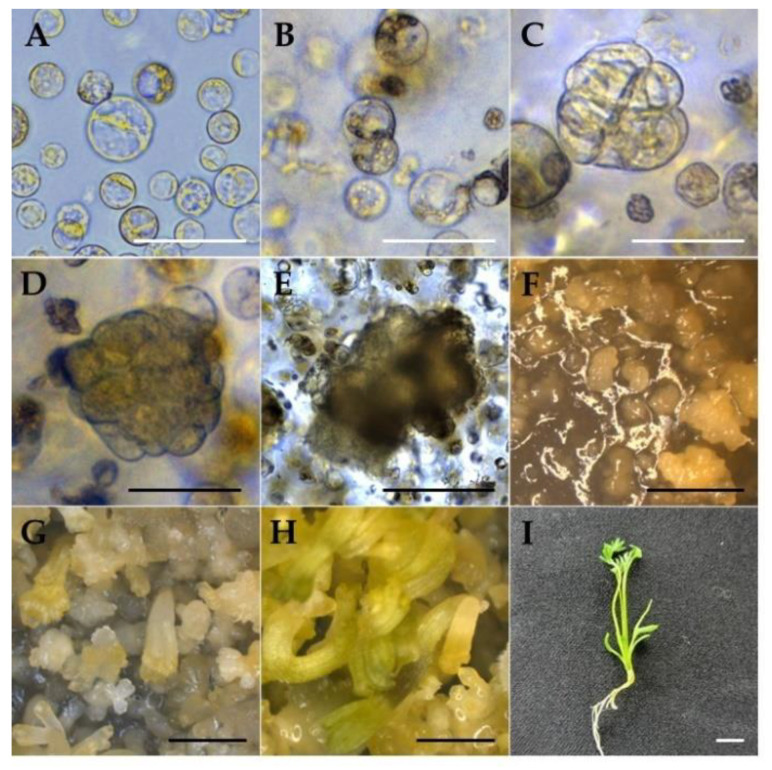
Different stages of EC-derived protoplast isolation and culture: (**A**) freshly isolated protoplast; (**B**) first cell division; (**C**) second cell division; (**D**) multi-cell division; (**E**) micro-callus formation; (**F**) EC and heart stage; (**G**) heart–torpedo stage (**H**); cotyledonary stage; (**I**) plant regeneration. (**A**–**C**: bar = 50 µm, (**D**,**E**) bar = 100 µm, (**F**–**I**) 2 mm.

**Figure 5 ijms-23-11556-f005:**
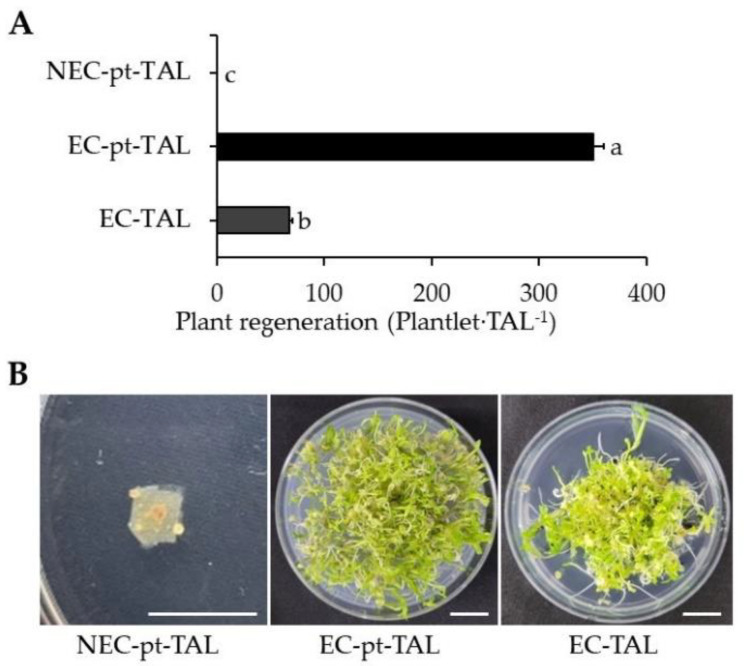
Plant regeneration from EC/NEC-pt-TAL and EC-TAL: (**A**) number of regenerated plants from TAL fragments; (**B**) images of plants regenerated from TAL fragments in the Petri dish. White scale bar = 1 cm.

**Figure 6 ijms-23-11556-f006:**
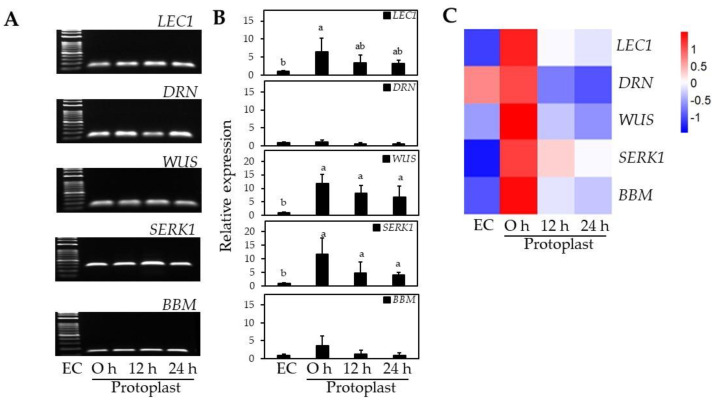
Expression of EC-related genes after protoplast isolation at 0, 12, and 24 h by RT-qPCR. (**A**) Electrophoresis of RT-qPCR products; (**B**) The relative expression levels of EC-related genes; (**C**) Heat map of relative expression data. Different lowercase letters indicate statistically significant differences (*p* < 0.05; Duncan’s multiple range test). Values are expressed as mean ± SD; the error bars represent at least three independent replicates.

## Data Availability

Not applicable.

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
