# Peer review of "Embryogenic Stem Cell Identity after Protoplast Isolation from Daucus carota and Recovery of Regeneration Ability through Protoplast Culture"

_ijms, 2022, doi:10.3390/ijms231911556_

Round 1

Reviewer 1 Report

The work is interesting. There are a few mistakes in the text. I would advise to read the text and try to make it smoother to read. There are many repetitions of the abbreviations, shortly one after another, what makes it difficult to understand. The addition of a few additional citations would also enhance the manuscript a lot. Discussion part is a little neglected. 

More comments in the text.

Author Response

The authors are thankful to anonymous reviewer for your valuable comments on the manuscript. We have revised the manuscript based on the suggestions of the reviewer and the revised part is marked in red on the manuscript. Following are specific changes made in the revised manuscript.

1, There are a few mistakes in the text. I would advise to read the text and try to make it smoother to read.

[R] All the corrections according to your comments were incorporated in the revised manuscript.

2. There are many repetitions of the abbreviations, shortly one after another, what makes it difficult to understand.

[R] Accoding to your suggestion, the abbreviations are used to a minimum, and we believe that it will  increase the readability of this manuscript.

3. The addition of a few additional citations would also enhance the manuscript a lot.

[R] According to the suggestion, the citations were added in the revised manuscript.

4. Discussion part is a little neglected. 

[R] Discussion section has been improved with more references than original version.

Reviewer 2 Report

Dear authors

I have read and revised your manuscript entitled: Maintenance of Embryogenic Stem Cell Identity after Protoplast Isolation from Daucus carota and Recovery of Re- generation Ability through Protoplast Culture

I have some comments about this manuscript.

Page 2:

You wrote:

In order to study the totipotency of ECs, many transcriptomic, metabolomic, and pro- teomic analyses have compared ECs with non-embryogenic callus (NEC) [14,15]. 

The references 14 and 15 does not mention nothing about transcriptomic, metabolomic and proteomic comparing EC vs Non-EC. Please provide at least 3 references, one of each, otherwise mention the most relevant, perhaps, transcriptomic.

Page 2:

You wrote:

Jha et al. [14] reported that various transcription factors (TFs) are regulated by seuen- tial and appropriate expression during the transition from vegetative to EC or the mainte- nance of EC identity. 

Not well written: By secuential

Mention at least 4 TFs involved in stem cell maintenance for somatic embryogenesis.

Page 2:

You wrote:

The process of protoplast isolation, protoplast culture, and plant regeneration are necessary for studies applying ECs. 

What do you mean?

Protoplast is indispensable to get ECs? Maybe you mean, that the protoplast culture is a good tool to understand some molecular mechanisms involved in somatic embryogenesis?

Page 7:

Please explain that:

EC is an embryogenic stem cell [35], is that true?

Results and Discusion

1.- You have stated that: SERK1, LEC1, BBM and Wuschel are involved in stem cell maintenance. Each of these TFs have different function during somatic embryo development. 

Please,  within results and discussion add:

a.- What TF is potentially involved in cell cycle regulation to induced protoplast cell divisions

b.- What TF was the one involved in cell wall regeneration

c.- What TF were involved in somatic embryo maturation

You can do a PPI network with high confidence (0.700) using STRING data base and the genes ID: LEC1, BBM, Wuschel and SERK1 using Arabidopsis thaliana genome. Apply enrichment analysis with 150 interactors and you will find the answers.

2.- How many embryogenic lines did you induced? and what parameter did you use to choose the line to isolate protoplasts?.

3.- The same question for the non-embryogenic line

4.- The term embryogenic callus is not very clear in all cases. The term callus means s a growing mass of unorganized plant parenchyma cells.  In your case, figure 4 F and G shows organized material resembling Proembryogenic masses (PEM). If you compared with the non-regeneration line, they are totally different.

Author Response

Response to Reviewer 2:

The authors are thankful to anonymous reviewer for your valuable comments on the manuscript. We have revised the manuscript based on the suggestions of the reviewer and the revised part is marked in blue on the manuscript. Following are specific changes made in the revised manuscript.

1. You wrote: In order to study the totipotency of ECs, many transcriptomic, metabolomic, and proteomic analyses have compared ECs with non-embryogenic callus (NEC) [14,15].

1) The references 14 and 15 does not mention nothing about transcriptomic, metabolomic and proteomic comparing EC vs non-EC. Please provide at least 3 references, one of each, otherwise mention the most relevant, perhaps, transcriptomic.

[R] Papers related to transcriptomic, metabolomic and proteomic comparing EC vs Non-EC were added to the references list with [14-18] in the revision..

  1. Guo, H.; Guo, H.; Zhang, L.; Tang, Z.; Yu, X.; Wu, J.; Zeng, F. Metabolome and Transcriptome Association Analysis Reveals Dynamic Regulation of Purine Metabolism and Flavonoid Synthesis in Transdifferentiation during Somatic Embryogenesis in Cotton. J. Mol. Sci. 2019, 20, 2070.
  2. Kumar, S.; Ruggles, A.; Logan, S.; Mazarakis, A.; Tyson, T.; Bates, M.; Grosse, C.; Reed, D.; Li, Z.; Grimwood, J.; Schmutz, J.; Saski, C. Comparative Transcriptomics of Non-Embryogenic and Embryogenic Callus in Semi-Recalcitrant and Non-Recalcitrant Upland Cotton Lines. Plants 2021, 10, 1775.
  3. Liu, B.; Shan, X.; Wu, Y.; Su, S.; Li, S.; Liu, H.; Han, J.; Yuan, Y. iTRAQ-Based Quantitative Proteomic Analysis of Embryogenic and Non-embryogenic Calli Derived from a Maize (Zea mays L.) Inbred Line Y423. J. Mol. Sci. 2018, 19, 4004.
  4. Shim, S.; Kim, H.K.; Bae, S.H.; Lee, H.; Lee, H.J.; Jung, Y.J.; Seo, P.J. Transcriptome comparison between pluripotent and non-pluripotent calli derived from mature rice seeds. Rep. 2020, 10, 21257.
  5. Wen, L.; Li, W.; Parris, S.; West, M.; Lawson, J.; Smathers, M.; Li, Z.; Jones, D.; Jin, S.; Saski, C.A. Transcriptomic profiles of non-embryogenic and embryogenic callus cells in a highly regenerative upland cotton line (Gossypium hirsutum L.). BMC Dev. Biol. 2020, 20, 25

2. You wrote: Jha et al. [14] reported that various transcription factors (TFs) are regulated by seuen- tial and appropriate expression during the transition from vegetative to EC or the mainte- nance of EC identity.

1) Not well written: By sequential, 2) Mention at least 4 TFs involved in stem cell maintenance for somatic embryogenesis.

[R] According to the comments, ‘sequential’ was removed, and TFs involved in stem cell maintenance for somatic embryogenesis were mentioned in the revised manuscript

  • Jha et al. [14] reported that various transcription factors (TFs), such as SOMATIC EMBRYOGENESIS RECEPTOR-LIKE KNASE (SERK), WUSCHEL (WUS), BABY BOOM (BBM) LEAFY COTYLEDON1/2 (LEC1/2), FUSCA3 (FUS3) and ABAINSENSITIVE3 (ABI3), are regulated by appropriate expression during the transition from vegetative to EC or the maintenance of EC identity.

3. You wrote: The process of protoplast isolation, protoplast culture, and plant regeneration are necessary for studies applying ECs. 1) What do you mean? Protoplast is indispensable to get ECs? Maybe you mean, that the protoplast culture is a good tool to understand some molecular mechanisms involved in somatic embryogenesis?

[R] Protoplast isolation, culture, and regeneration processes are useful methods for understanding molecular mechanisms to study the stem cell characteristics of ECs. This is mentioned in the revision.

4. EC is an embryogenic stem cell [35], is that true?

[R] EC has a morphological and functional characteristics of stem cells, and many of references [2,38,39,40] supports of it. The concept of embryogenic stem cells was described by Verdeil et al. [2], and EC of animal has been also considered as pluripotent stem cells for a long time. According to the reviewer's comments, these are clarified in the uploaded revision.

5. You have stated that: SERK1, LEC1, BBM and Wuschel are involved in stem cell maintenance. Each of these TFs have different function during somatic embryo development. Please, within results and discussion add: What TF is potentially involved in cell cycle regulation to induced protoplast cell divisions, What TF was the one involved in cell wall regeneration, What TF were involved in somatic embryo maturation. You can do a PPI network with high confidence (0.700) using STRING data base and the genes ID: LEC1, BBM, Wuschel and SERK1 using Arabidopsis thaliana genome. Apply enrichment analysis with 150 interactors and you will find the answers.

[R] Since SERK1, LEC1, BBM and WUSCHEL are involved in cell proliferation and embryo development, it is thought that four TFs are involved in cell division. In addition, BBM is known as a TF involved in cell wall regeneration, and LEC1, SERK1 and BBM are known to be involved in somatic embryonic maturation during somatic embryogenesis. The results and discussion sections have been revised on manuscript.

6. How many embryogenic lines did you induced? and what parameter did you use to choose the line to isolate protoplasts?

[R] More than 20 embryogenic calli were induced from carrot seeds, and among them, vigorously proliferated ECs was selected and maintained as a EC lines. This information was added in the Materials and Methods sections.

7. The same question for the non-embryogenic line

[R] More than 50 masses of NECs were induced from the hypocotyl of carrots, and well proliferated NEC were selected and maintained as a line for protoplast isolation. This information was added in the Materials and Methods sections.

8. The term embryogenic callus is not very clear in all cases. The term callus means a growing mass of unorganized plant parenchyma cells.

[R] In this manuscript, ‘embryogenic callus’ indicated the mass of proliferated embryogenic cells on solid medium, and ‘embryogenic cells’ indicated fine and separated single-multi cells. We corrected the manuscript by using this term consistently through the manuscript.

9. In your case, figure 4 F and G shows organized material resembling Proembryogenic masses (PEM). If you compared with the non-regeneration line, they are totally different. 

[R] We agreed to your comment, so Figure 4 has been revised in the manuscript.
